# Nutrition in the First Week after Stroke Is Associated with Discharge to Home

**DOI:** 10.3390/nu13030943

**Published:** 2021-03-15

**Authors:** Yoichi Sato, Yoshihiro Yoshimura, Takafumi Abe

**Affiliations:** 1Department of Rehabilitation, Uonuma Kikan Hospital, Minamiuonuma 949-7302, Japan; yoichi3041@gmail.com (Y.S.); t.abe.pt@gmail.com (T.A.); 2Center for Sarcopenia and Malnutrition Research, Kumamoto Rehabilitation Hospital, Kumamoto 869-1106, Japan

**Keywords:** energy intake, home-discharge, activity of daily living, stroke

## Abstract

Malnutrition is associated with poor clinical outcomes in stroke patients. The effect of early nutritional intake after admission on home discharge is unclear. We evaluated the impact of energy intake in the first week of hospitalization of acute stroke patients on home discharge and activities of daily living (ADL). A retrospective cohort study was conducted with 201 stroke patients admitted to an acute care hospital in Japan. The energy and protein intake during the first week were evaluated. Multivariate models were used to estimate variables related to discharge destination and ADL at discharge. The cut-off point of nutritional intake for determining the discharge destination was evaluated using the receiver operating characteristic curve. Out of 163 patients included in the analysis, 89 (54.6%) and 74 (45.4%) were discharged home and elsewhere, respectively. Those discharged home had higher energy and protein intake than those discharged elsewhere. In multiple regression analysis, energy intake was independently associated with ADL at discharge and home discharge (odds ratio 1.146). Those with energy intake >20.7 kcal/kg/day had higher ADL at discharge and more patients discharged home than those with energy intake <20.7 kcal/kg/day. Energy intake during the first week affected home discharge in acute stroke patients.

## 1. Introduction

Malnutrition in post-stroke patients results in increased mortality, complications, and poor functional prognosis [1,2,3]. Malnutrition is associated with dysphagia and impaired consciousness [3,4]. Prolonged starvation uses skeletal muscle for energy and causes muscle loss and muscle dysfunction [5], leading to the onset or exacerbation of sarcopenia, which is frequently observed in post-stroke patients, and stroke patients with sarcopenia have poor improvement in physical and swallowing functions [6,7].

Energy intake after admission is associated with functional improvement at discharge. In convalescent rehabilitation wards in Japan, energy intake during the first week after admission is associated with activities of daily living (ADL) at discharge in post-stroke patients [3]. In the acute phase, 26.4% of stroke patients are malnourished one week after admission, and high energy intake in the first week improves ADL at discharge [1,3].

Sex, stroke category, severity, level of ADL, and nutritional status at admission have been reported as factors predicting home discharge in acute stroke patients [8,9,10,11]. However, the effect of early nutritional intake after admission on home discharge is unclear. In addition, both energy and protein intake may affect the nutritional status of acute stroke patients [1]; however, the effect of protein intake on the discharge destination is unknown.

Protein-energy malnutrition worsens ADL after 30 days of admission in acute stroke patients [1] and may affect their discharge destination, because high ADL level is associated with home discharge [8,11]. Therefore, in this study, we evaluated the impact of energy and protein intake during the first week of admission on home discharge and ADL in acute stroke patients. The results suggest that early and active nutritional support in the acute phase may affect functional prognosis and post-discharge destination.

## 2. Materials and Methods

### 2.1. Participants and Setting

We conducted a single-center retrospective cohort study at a 454-bed acute care hospital in Niigata, Japan. We enrolled 201 consecutive patients who were admitted between May 2020 and January 2021 with cerebral infarction and cerebral hemorrhage within 48 h after onset. The presence of stroke was confirmed in all enrolled patients using computed tomography or magnetic resonance imaging [12]. The exclusion criteria were (1) missing data, (2) altered consciousness, (3) pacemaker implantation, (4) admission for diseases other than stroke, and (5) death. The observation period was the duration of hospitalization (from the date of admission to the date of discharge).

During the study period, 201 stroke patients were registered (Figure 1). Based on the exclusion criteria, 38 patients were excluded: missing data (*n* = 18), altered consciousness (*n* = 10), pacemaker implantation (*n* = 3), admission of diseases other than stroke (*n* = 5), and death during hospitalization (*n* = 2). Finally, 163 participants (mean age 75.2 ± 12.6 years; 36.8% women) were analyzed.

The rehabilitation program (up to 3 h/day), aimed at improving endurance, ADL training, and dysphagia rehabilitation, was tailored to accommodate the functional abilities and disabilities of the individual patient and included paralyzed-limb facilitation, range-of-motion exercises, basic movement training (mainly for the legs), walking training, resistance training, and aerobic exercises using an ergometer [13,14]. Rehabilitation therapy was performed in a general way according to the patient’s functional abilities and disabilities.

### 2.2. Data Collection

Basic information was recorded at admission, such as age, sex, body mass index (BMI), stroke type, stroke severity (based on National Institutes of Health Stroke Scale; NIHSS), days from onset, and other laboratory data (serum concentration of albumin and hemoglobin). Nutritional status was assessed using the Geriatric Nutritional Risk Index (GNRI) [15,16], which was calculated from the serum albumin concentration and body weight using the following equation: GNRI = [14.89 × albumin concentration (g/dL)] + [41.7 × (actual body weight/ideal body weight)]. Ideal body weight was defined as a BMI of 22.0 kg/m^2^. Skeletal muscle mass and handgrip strength were measured within 5 days of admission. The grip strength was measured using a Smedley hand dynamometer, and the maximum value out of two measurements on each side was considered. In case of paralysis, the value of the hand without paralysis was considered. If both hands could not be measured due to coma, they were excluded from the handgrip strength results. Skeletal muscle mass was assessed using bioelectrical impedance analysis using InBody S10 (InBody, Tokyo, Japan), and the skeletal muscle index (SMI) was calculated based on skeletal muscle mass [17]. Swallowing function was assessed using the Functional Oral Intake Scale (FOIS) [18]. It was determined based on the dietary pattern after evaluation by a speech therapist within 3 days of admission.

### 2.3. Nutrition Intake

Based on a previous study [19], we reviewed diet records to quantify the mean daily nutritional intake during the first week after admission to the acute care hospital. Oral intake was measured by the experienced nurses using visual estimation. Visual estimation is commonly used in hospitals to evaluate food intake through the estimation of plate waste [14], and it comprises visually estimating the food present before and after the plate is provided to the patient. After the patient finished the meal, the nurse would register how much food the patient has taken in on an 11 point scale of 0–10. The oral nutritional intake was calculated from the amounts of calories and protein provided in the diet and amount of dietary intake. Energy and protein content during enteral and parenteral nutrition were collected from medical records. Energy and protein intake was defined as the mean value of nutritional intake in the first week divided by body weight.

### 2.4. Outcome Measurement

The primary outcome was the discharge destination from the acute care hospital, and this was categorized as home and others (convalescent rehabilitation hospital or nursing facility).

The secondary outcome was ADL, which was assessed using the Functional Independence Measure (FIM). The FIM score rates 13 motor and 5 cognitive activities on a scale of 1 (complete dependence) to 7 (complete independence). The total FIM score ranges from 18 to 126 points. A high FIM score indicates high activities of daily living [7]. The FIM gain was defined as the FIM at discharge minus the FIM at admission.

### 2.5. Sample Size Calculation and Statistical Analysis

We divided the patients into two groups: those discharged to home and those discharged to other facilities. To compare nutritional intake between the two groups, the null hypothesis was rejected with a sample size of at least 64 participants in each group with 0.5 effect size, 0.8 detection power, and 0.05 alpha error.

The results are reported as mean (standard deviation, SD) for parametric data, as median (25th to 75th percentile or interquartile range, IQR) for nonparametric data, and as number (%) for categorical data. The unpaired t test, Mann–Whitney U test, and Chi-square test were performed for comparison between the two groups. Multiple logistic regression analysis was used to determine whether nutritional intake was independently associated with home-discharge (primary outcome) from the acute care hospital. Based on previous studies [10,20,21,22,23], we selected age, sex, stroke category, NIHSS score, length of hospital stay, nutritional status, SMI, handgrip strength, swallowing function, FIM score at admission, FIM eating at discharge, FIM gain, paralysis (lower limbs), and rehabilitation time as covariates. If energy intake or protein intake had a significant effect on home discharge, the cut-off point of nutritional intake for determining the discharge destination was evaluated using the receiver operating characteristic (ROC) curve. We used the value at which the Youden Index was the highest as a criterion for determining the cut-off point. The patient characteristics were compared by dividing them into two groups at the cut-off point. Multiple linear regression analysis was used to determine whether nutritional intake was independently associated with FIM at discharge (secondary outcome). Using the same covariates (excluding FIM eating at discharge, FIM gain) as in the logistic regression analysis, we investigated the effects of energy and protein intake on ADL at discharge. Multicollinearity was assessed using the variance inflation factor (VIF): VIF value between 1 and 10 was considered as the absence of multicollinearity. All analyses were performed using SPSS version 21 (IBM, Armonk, NY, USA). *p* < 0.05 was considered statistically significant.

### 2.6. Ethics

This study was approved by the Institutional Review Board of the study center (02–024). Written informed consent could not be obtained because of the constraints imposed by the retrospective study design, although the participants could withdraw from this study at any time by using an opt-out procedure. This study was conducted in accordance with the Declaration of Helsinki and ethical guidelines for medical and health research involving human subjects.

## 3. Results

The baseline characteristics of the enrolled participants are summarized in Table 1. Of these, 89 (54.6%) and 74 (45.4%) were discharged home and elsewhere, respectively. The median FIM score at admission was 58 (IQR: 32–83), suggesting that a large number of patients were physically dependent at baseline. The NIHSS score at admission was 5 (2–9) points. The median length of hospital stay was 19 (11–28) days. The median rehabilitation time was 60.4 (45.8–76.0) min/day. The home discharge group had significantly younger patients (*p <* 0.001), more male patients (*p* = 0.011), and patients with lower severity of stroke, milder motor paralysis, and shorter length of hospital stay than the other discharge group (all *p <* 0.001). Moreover, SMI, handgrip strength, and swallowing function (FOIS) at admission were significantly higher in the home discharge group than the other discharge group (all *p <* 0.001). The rehabilitation time was significantly longer in the other discharge group than in the home discharge group (*p* = 0.022). The FIM score at admission in the other discharge group was 32 (20–51), and many patients required assistance for ADL. In contrast, the score in the home discharge group was 80 (66–93), and some patients showed early onset independence. The FIM eating at discharge was higher in the home discharge group (*p <* 0.001), but FIM gain was higher in the other discharge group (*p* = 0.001). The energy intake in the home discharge group was 23.5 (16.7–26.6) kcal/kg/day which was significantly higher than that in the other discharge group (12.4 (9.3–18.4) kcal/kg/day). Similarly, the protein intake in the home discharge group was 0.9 (0.8–1.1) g/kg/day, which was significantly higher than that in the other discharge group (0.7 (0.5–0.9) g/kg/day).

Table 2 shows the results of the multivariate analysis according to the discharge destination and FIM at discharge. There was no multicollinearity between the variables. Multiple logistic regression analysis showed that energy intake (odds ratio (OR) = 1.146, 95% confidence interval (CI) = 1.029–1.276, *p* = 0.013), FOIS (OR = 1.450, 95% CI = 1.036–2.544, *p* = 0.036), FIM at admission (OR = 1.039, 95% CI = 1.003–1.075, *p* = 0.033), and FIM eating at discharge (OR = 1.651, 95% CI = 1.952–2.063, *p* = 0.045) were significantly associated with home discharge. The Hosmer–Lemeshow test was *p* = 0.887 with high prediction accuracy. The discriminative predictive value of this logistic regression analysis was 87.5%. This result indicates that high energy intake is an independent predictor of home discharge. The multiple linear regression analysis shows that energy intake (β = 0.131, *p* = 0.025), length of stay (β = 0.108, *p* = 0.041), SMI (β = 0.164, *p* = 0.019), Handgrip strength (β = 0.166, *p* = 0.028), FIM at admission (β = 0.349, *p* < 0.001), and Brunnstrom recovery stage (BRS)-lower limb (β = 0.123, *p* = 0.049) were positively associated and NIHSS (β = −0.164, *p* = 0.020) was negatively and independently associated with FIM score at discharge (Adjusted R2 = 0.799). Protein intake was not a significant variable for home discharge or FIM score at discharge.

The ROC curve of energy intake for the determination of home discharge is shown in Figure 2. The area under the curve (95% CI) for the energy intake was 0.795 (0.726–0.864). Energy intake was a significant predictive variable (*p* < 0.001). The cut-off point of energy intake for determining home discharge was 20.7 kcal/kg/day (sensitivity, 0.618; specificity, 0.838). Table 3 shows the evaluation of patients in each group divided according to the cut-off point: those above the cut-off point (High Group) and those below the cut-off point (Low Group). There was no significant difference in the GNRI between the two groups. The length of hospital stay was significantly shorter in the High Group than in the Low Group (*p* = 0.012). The FOIS and FIM scores at discharge and home discharge rate were significantly higher in the High Group than in the Low Group (all *p <* 0.001).

## 4. Discussion

In this study, we reported the relationship between nutritional intake and discharge destination in the early phase of hospitalization in acute stroke patients. We report that in acute stroke patients: (1) energy intake during the first week of admission was independently associated with home discharge; (2) 20.7 kcal/kg/day might be the cut-off point for predicting home discharge; (3) protein intake was not related to home discharge and the level of ADL at discharge.

Energy intake in the early phase after onset in acute stroke patients is associated with functional prognosis. In a previous study on acute stroke patients, high energy intake in the early phase after onset resulted in significant improvements in ADL [24]. Kokura et al. measured energy intake for one week after admission and compared it with the basal energy expenditure calculated using the Harris–Benedict equation [19]. They reported that the energy-sufficient group had a higher FIM gain at discharge than the energy-deficient group, and many patients had improved nutritional status. However, only 39.1% of patients were energy-sufficient, indicating that many acute stroke patients had inadequate energy intake [19]. Malnutrition is present in 26.4% and 35% of acute stroke patients in the first and second week after admission, respectively [1]. In this study, we did not assess body weight at discharge, so we did not know how many patients lost weight due to energy deficient. Aggressive nutritional support in the early phase after onset, when there is a high risk of malnutrition, may lead to significant improvements in ADL. Moreover, it has been reported that a high level of ADL was associated with high home discharge rate [10,11,21,25]. This evidence supports our findings. Furthermore, consistent with previous studies, this study reports that good swallowing function was independently associated with home discharge [8,9]. Therefore, early detection of dysphagia is essential in the nutritional management of post-stroke patients, including dietary modification.

Energy intake predicting home discharge was 20.7 kcal/kg/day in this study population. High energy intake patients above this cut-off point exhibited higher ADL recovery and home discharge rates than those below the cut-off point. In a previous study on acute stroke patients, the energy sufficient group (median: 21.5 kcal/kg/day) had a higher ADL recovery than the deficient group (median: 10.3 kcal/kg/day) [18]. Acute stroke patients with inadequate energy intake (median: 11.6 kcal/kg/day) reported more loss of quadriceps muscle on the non-paralyzed side than those with adequate energy intake (median: 24.0 kcal/kg/day) [19]. Therefore, the cut-off point of this study supports the findings of previous studies. Contrarily, in convalescent rehabilitation hospitals, post-stroke patients with severe malnutrition (GNRI < 82) had an average energy intake of 26.9 kcal/kg/day [16]. However, the study reported that this energy intake may not be sufficient to improve weight loss in patients with severe malnutrition [16]. Thus, it is necessary to provide appropriate energy intake depending on the phase of the disease and the status of malnutrition.

In this study, protein intake did not affect the discharge destination or FIM score at discharge. There are two possible reasons for this. First, the participants of this study included patients who were provided with a protein-restricted diet due to renal dysfunction. In the elderly, 1.0–1.2 g/kg/day of protein intake is recommended to maintain or improve lean body mass [26,27]. However, it is possible that some patients were provided with a diet with less protein content than recommended due to renal dysfunction. Second, protein quality may affect protein intake. Randomized controlled trials in Japan showed that the combination of “leucine-rich” or “branched-chain amino acid intake” and “resistance training” improved muscle mass, strength, and physical function in stroke patients with sarcopenia [28,29,30]. Therefore, the intake of high-quality amino acids, but not total protein content, may predict the discharge destination and functional prognosis of stroke patients.

The present study has several limitations. First, because it was a retrospective cohort study in a single acute care hospital, we cannot generalize the results, eliminate potential confounders, or prove a causal relationship between energy intake and discharge destination. Second, nurses visually assessed nutritional intake, the main parameter, which may have resulted in measurement errors. Third, the physical functions of patients with impaired consciousness could not be accurately assessed. For example, the handgrip strength of coma patients was excluded. Fourth, we did not record the foods that family members brought from the outside. It is possible that they are taking in more nutrients than we have assessed. These limitations can be overcome by conducting a prospective cohort study at multiple centers. In the future, it will be necessary to examine the relationship between energy intake and discharge destination in interventional studies.

## 5. Conclusions

Energy intake during the first week of admission is associated with discharge destination and ADL at discharge in acute stroke patients. This finding suggests that early detection of nutrition-related factors, such as dysphagia and aggressive nutritional support, may enhance home discharge in this setting.

## Figures and Tables

**Figure 1 nutrients-13-00943-f001:**
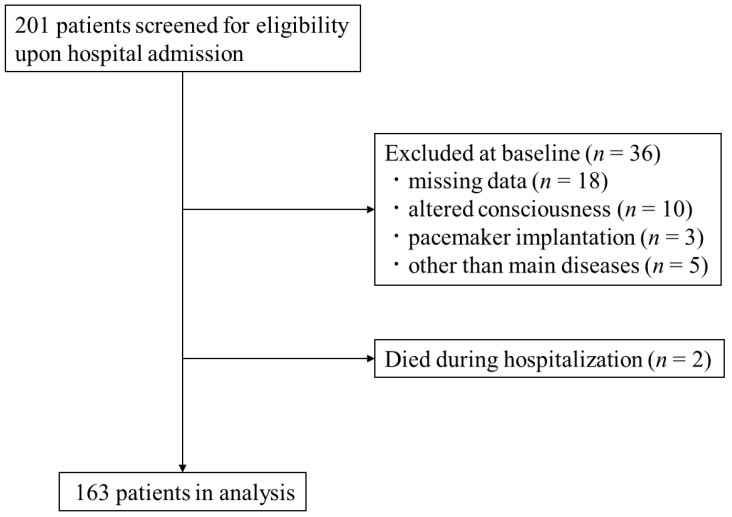
Flowchart of the study population.

**Figure 2 nutrients-13-00943-f002:**
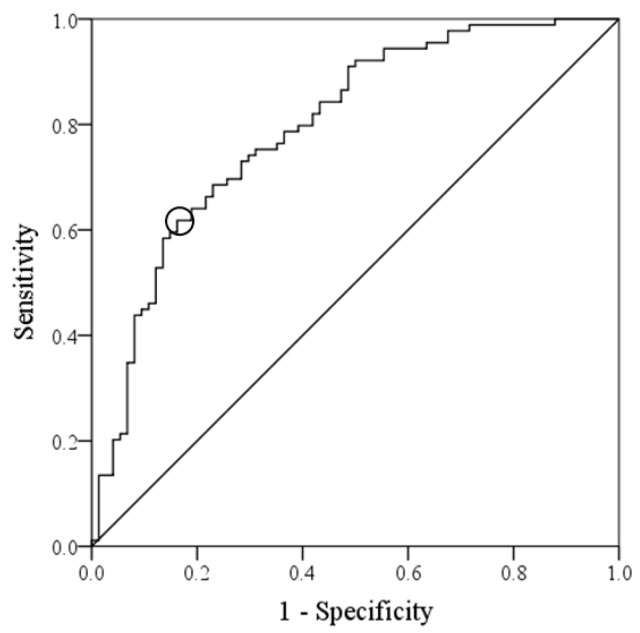
Receiver-operating characteristic curve analysis for home-discharge. The cut-off points of energy intake for determining home discharge rate were 20.7 kcal/kg/day (sensitivity 0.618, specificity 0.838, area under the curve 0.795, 95% confidence interval 0.726–0.864, *p* < 0.001); the circle indicates cut-off point.

**Table 1 nutrients-13-00943-t001:** Comparison of patient characteristics between the two groups by discharge destination.

	Total (*n* = 163)	Home (*n* = 89)	Other (*n* = 74)	*p*–Value
Age (year)	75.2 (12.6)	71.4 (12.6)	78.4 (11.8)	<0.001
Sex (male/female)	103/60	64/25	39/35	0.011
Body mass index (kg/m^2^)	22.8 (3.9)	23.1 (3.9)	22.3 (3.9)	0.205
NIHSS score	5 (2–9)	2 (1–5)	9 (6–14)	<0.001
Stroke type (infarct/hemorrhage)	129/34	75/14	54/20	0.077
Comorbidity (%)				
Hypertension	109 (66.9)	61 (68.5)	48 (64.9)	0.547
Diabetes	54 (33.1)	26 (29.2)	28 (37.8)	0.244
Previous stroke	19 (11.7)	7 (7.9)	12 (16.2)	0.098
Atrial fibrillation	50 (30.7)	23 (25.8)	27 (36.5)	0.142
Side of lesion (right/left/both)	65/89/9	38/45/6	27/44/3	0.401
BRS				
Upper limb	5 (3–6)	6 (5–6)	3 (2–5)	<0.001
Hand-finger	5 (3–6)	5 (5–6)	3 (2–5)	<0.001
Lower limb	5 (4–6)	6 (5–6)	4 (2–5)	<0.001
Days from onset (day)	0 (0–0)	0 (0–0)	0 (0–0)	0.989
Length of hospital stay (day)	19 (11–28)	14 (10–19)	27 (20–34)	<0.001
Laboratory data				
Albumin (g/dL)	4.0 (0.5)	4.1 (0.5)	30.9 (00.5)	0.079
Hemoglobin (g/dL)	13.3 (2.1)	13.5 (2.1)	130.1 (20.2)	0.268
GNRI	103.3 (95.2–108.9)	104.7 (96.7–110.5)	101.0 (93.6–107.8)	0.275
SMI (kg/m^2^)	7.0 (6.0–8.1)	7.2 (6.3–8.6)	6.4 (5.4–7.6)	<0.001
Handgrip strength (kg)	22.9 (14.0–30.2)	25.0 (18.5–33.0)	15.2 (8.6–20.0)	<0.001
FOIS	5 (2–6)	6 (5–7)	2 (1–5)	<0.001
Rehabilitation time (min/day)	60.4 (45.8–76.0)	54.8 (37.9–74.5)	64.0 (52.5–77.4)	0.022
FIM at admission	58 (32–83)	80 (66–93)	32 (20–51)	<0.001
FIM eating at discharge	7 (5–7)	7 (7–7)	4 (2–6)	<0.001
FIM gain	24 (11–40)	22 (11–40)	28 (11–39)	0.001
Energy intake (kcal/kg/day)	18.3 (12.1–24.7)	23.5 (16.7–26.6)	12.4 (9.3–18.4)	<0.001
Protein intake (g/kg/day)	0.9 (0.6–1.0)	0.9 (0.8–1.1)	0.7 (0.5–0.9)	<0.001

Mean (SD) or median (IQR) or subjects (%); BRS: Brunnstrom stage, GNRI: Geriatric Nutritional Risk Index, SMI: Skeletal Muscle Mass Index; FOIS: Functional Oral Intake Scale, FIM: Functional Independence Measure; Handgrip strength: Home (*n* = 78), Other (*n* = 42).

**Table 2 nutrients-13-00943-t002:** Multivariate analysis for home discharge and FIM at discharge.

	Home–Discharge ^#1^		FIM at Discharge ^#2^	
	OR (95% CI)	*p*–Value	β	*p*-Value
Age	0.966 (0.903–1.034)	0.323	−0.065	0.131
Gender (male)	1.160 (0.214–6.293)	0.864	−0.048	0.302
Stroke type(infarction)	0.466 (0.079–2.729)	0.397	−0.084	0.109
NIHSS	0.836 (0.637–1.098)	0.198	−0.164	0.020
Length of stay	0.974 (0.937–1.012)	0.180	0.108	0.041
GNRI	1.012 (0.977–1.048)	0.514	0.060	0.205
SMI	1.137 (0.551–2.345)	0.729	0.164	0.019
Handgrip strength	1.057 (0.922–1.212)	0.425	0.166	0.028
FOIS	1.450 (1.036–2.544)	0.036	0.070	0.278
FIM at admission	1.039 (1.003–1.075)	0.033	0.349	<0.001
FIM eating at discharge	1.651 (1.952–2.063)	0.045		
FIM gain	1.042 (0.979–1.109)	0.193		
BRS-lower limb	0.819 (0.432–1.549)	0.539	0.123	0.049
Rehabilitation therapy	0.968 (0.941–1.083)	0.074	0.109	0.121
Energy intake	1.146 (1.029–1.276)	0.013	0.131	0.025
Protein intake	0.104 (0.006–1.739)	0.115	0.063	0.264

^#1^ Multiple logistic regression analysis, ^#2^ Multiple linear regression analysis; NIHSS: National Institutes of Health Stroke Scale, GNRI: Geriatric Nutritional Risk Index; SMI: Skeletal Muscle Mass Index, FOIS: Functional Oral Intake Scale; FIM: Functional Independence Measure, BRS: Brunnstrom stage.

**Table 3 nutrients-13-00943-t003:** Comparison between two groups divided by cut-off point of energy intake.

	High Group (*n* = 67)	Low Group (*n* = 96)	*p*-Value
Length of hospital stay (day)	14 (10–24)	22 (16–31)	0.012
GNRI	103.3 (94.7–108.3)	102.9 (95.1–109.4)	0.270
FOIS	6 (5–7)	4 (1–5)	<0.001
FIM at discharge	122 (97–126)	78 (40–111)	<0.001
Discharge to home (%)	55 (82.1)	34 (35.4)	<0.001

Median (IQR) or subjects (%); GNRI: Geriatric Nutritional Risk Index, FOIS: Functional Oral Intake Scale; FIM: Functional Independence Measure; two groups divided by cut-off point of energy intake (20.7 kcal/kg/day).

## Data Availability

The data are not publicly available owing to opt out restrictions. Data sharing is not applicable.

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
