# Peer review of "Nutrition in the First Week after Stroke Is Associated with Discharge to Home"

_nutrients, 2021, doi:10.3390/nu13030943_

Round 1

Reviewer 1 Report

This is a well-designed study and a well written manuscript.  The quality of the English is excellent, the statistical analysis is sound and the conclusions reached by the authors reflect the results of the study.

The primary weakness is the method of measurement used to estimate the food intake of each patient- and the authors have acknowledged this fact.  I do, however, have one minor criticism.  While the authors have tested for collinearity between predictors, a secondary consideration that they have not addressed thoroughly is a possible interaction between each patient's ability to feed themselves (and hence their nutritional intake) and discharge destination.  In Australia, a patient who needed assistance to eat at the time of discharge would almost certainly not be discharged home.  These patients might also be expected to have poorer nutritional intake.  While the authors have corrected for the overall FIM score at admission, the study could be improved by also including the specific "Eating" score of the FIM at time of discharge in the regression model, and testing for interactions between this score and nutritional intake with regard to discharge destination.

Author Response

Dear editors and reviewers in Nutrients. Thank you for your positive comments in the previous review, which really helped us improve our manuscript. We have made additional changes in this review on the comments. Our changes have marked in red in the revised manuscript.

REVIEWER #1:

Comment 1: The primary weakness is the method of measurement used to estimate the food intake of each patient- and the authors have acknowledged this fact.  I do, however, have one minor criticism.  While the authors have tested for collinearity between predictors, a secondary consideration that they have not addressed thoroughly is a possible interaction between each patient's ability to feed themselves (and hence their nutritional intake) and discharge destination.  In Australia, a patient who needed assistance to eat at the time of discharge would almost certainly not be discharged home.  These patients might also be expected to have poorer nutritional intake.  While the authors have corrected for the overall FIM score at admission, the study could be improved by also including the specific "Eating" score of the FIM at time of discharge in the regression model, and testing for interactions between this score and nutritional intake with regard to discharge destination.

(Response)

We appreciate your positive comments and feedback very much. I agree with your concern that the nutritional intake of patients who need assistance with eating might be reduced. To address this concern and improve the quality of the study, the “FIM-eating score at hospital discharge” was added to the multivariate adjustment for outcomes. Accordingly, we revised the relevant sentences and Tables as follows.

(Change)

(“2.5. Sample size calculation and statistical analysis” section in Materials and Methods) “. . . Based on previous studies [10, 20, 21], we selected age, sex, stroke category, NIHSS score, length of hospital stay, nutritional status, SMI, handgrip strength, swallowing function, FIM score at admission, FIM eating at discharge, FIM gain, paralysis (lower limbs), and rehabilitation time as covariates. If energy intake or protein intake had a significant effect on home discharge, the cut-off point of nutritional intake for determining the discharge destination was evaluated using the receiver operating characteristic (ROC) curve. We used the value at which the Youden Index was the highest as a criterion for determining the cut-off point. The patient characteristics were compared by dividing them into two groups at the cut-off point. Multiple linear regression analysis was used to determine whether nutritional intake was independently associated with FIM at discharge (secondary outcome). Using the same covariates (excluding FIM eating at discharge, FIM gain) as in the logistic regression analysis, we investigated the effects of energy and protein intake on ADL at discharge. Multi-collinearity was assessed using the variance inflation factor (VIF): VIF value between 1 and 10 was considered as the absence of multicollinearity. All analyses were performed using SPSS version 21 (IBM, Armonk, NY, USA). p<0.05 was considered statistically significant.

(First paragraph in “3. Results” section) “. . . The FIM score at admission in the other discharge group was 32 (20–51), and many patients required assistance for ADL. In contrast, the score in the home discharge group was 80 (66–93), and some patients showed early onset independence. The FIM eating at dis-charge was higher in the home discharge group (p <0.001), but FIM gain was higher in the other discharge group (p = 0.001). The energy intake in the home discharge group was 23.5 (16.7–26.6) kcal/kg/day which was significantly higher than that in the other discharge group (12.4 [9.3–18.4] kcal/kg/day). Similarly, the protein intake in the home discharge group was 0.9 (0.8–1.1) g/kg/day, which was significantly higher than that in the other discharge group (0.7 [0.5–0.9] g/kg/day).”

(Second paragraph in “3. Results” section) “Table 2 shows the results of the multivariate analysis according to the discharge destination and FIM at discharge. There was no multicollinearity between the variables. Multiple logistic regression analysis showed that energy intake (odds ratio [OR] = 1.146, 95% confidence interval [CI] = 1.029–1.276, p=0.013), FOIS (OR = 1.450, 95% CI = 1.036–2.544, p=0.036), FIM at admission (OR = 1.039, 95% CI = 1.003–1.075, p=0.033) and FIM eating at discharge (OR = 1.651, 95% CI = 1.952–2.063, p=0.045) were significantly associated with home discharge. The Hosmer-Lemeshow test was p = 0.887 with high prediction accuracy. The discriminative predictive value of this logistic regression analysis was 87.5%. This result indicates that high energy intake is an independent predictor of home discharge. The multiple linear regression analysis shows that energy intake (β = 0.131, p=0.025), length of stay (β = 0.108, p=0.041), SMI (β = 0.164, p=0.019), Handgrip strength (β = 0.166, p=0.028), FIM at admission (β = 0.349, p<0.001) and Brunnstrom recovery stage (BRS)-lower limb (β = 0.123, p=0.049) were positively associated and NIHSS (β = -0.164, p=0.020) was negatively and independently associated with FIM score at discharge (Adjusted R2 = 0.799). Protein intake was not a significant variable for home discharge or FIM score at discharge.

Table 1 and Table 2 have been revised according to the updated results.

Reviewer 2 Report

This study aimed to examine the influence of nutritional intake during the first week of admission following a stroke on discharge disposition. Though this study significantly adds to the stroke literature, as little is currently documented regarding dietary intake and acute stroke recovery, several clarifications may aid this manuscript.

  • Please clarify if the rehabilitation program just usual care therapy?
  • Please clarify the reasons that handgrip could not be measured as it seems inaccurate to represent a lack of measurement as 0 kg in some instances (i.e. if unable to perform due to IV placement). This seems more accurately represented as missing data. Did these situations differ between the 2 groups?
  • It should be mentioned as a limitation that family members could bring food in from the outside.
  • How were nurses trained to evaluate plate waste?
  • Please clarify what is meant by the nurses would measure “grain and dishes intake”?
  • Since rehabilitation time was longer in the other discharge group, was energy expenditure during this activity considered when assessing nutritional needs?
  • Assessment in changes in some of these outcomes over time should be included if available (i.e. changes in body weight, FIM, strength, etc). Did caloric inadequacy lead to weight loss in either group?
  • How did actual intake compare to that prescribed by the dietitian? Did each patient see a dietitian at admission?

Author Response

RESPONSE TO REVIEWER #2:

Comment 1: Please clarify if the rehabilitation program just usual care therapy?

(Response)

We appreciate your positive comments and feedback very much. The rehabilitation carried out in this study was not unique. The following statements have been added to the text to aid the reader's understanding.

(Change)

(Third paragraph of “2. Materials and Methods” section) “The rehabilitation program (up to 3 h/day), aimed at improving endurance, ADL training, and dysphagia rehabilitation, was tailored to accommodate the functional abilities and disabilities of the individual patient and included paralyzed-limb facilitation, range‐of‐motion exercises, basic movement training (mainly for the legs), walking training, resistance training, and aerobic exercises using an ergometer [13, 14]. Rehabilitation therapy was performed in a general way according to the patient's functional abilities and disabilities.

Comment 2: Please clarify the reasons that handgrip could not be measured as it seems inaccurate to represent a lack of measurement as 0 kg in some instances (i.e. if unable to perform due to IV placement). This seems more accurately represented as missing data. Did these situations differ between the 2 groups?

(Response)

The reason why handgrip strength could not be measured was that the patient was in coma and communication was difficult. We consider that defining a patient who could not measure for handgrip strength as "0.0 kg" is an underestimation of physical function. Therefore, we agree that the unmeasured handgrip strength value should be defined as a missing data. The following statements have been added to the text to aid the reader's understanding. Table 1 and Table 2 have been revised according to the updated results. 11 patients in the home discharge group and 32 patients in the other discharged group could not measure their handgrip strength.

        (Change)

(“2.2. Data collection” section in Materials and Methods) “. . . The grip strength was measured using a Smedley hand dynamometer, and the maximum value out of two measurements on each side was considered. In case of paralysis, the value of the hand without paralysis was considered. If both hands could not be measured due to coma, they were excluded from the handgrip strength results.

(Fifth paragraph of “4. Discussion” section) “The present study has several limitations. First, because it was a retrospective cohort study in a single acute care hospital, we cannot generalize the results, eliminate potential confounders, or prove a causal relationship between energy intake and discharge destination. Second, nurses visually assessed nutritional intake, the main parameter, which may have resulted in measurement errors. Third, the physical functions of patients with impaired consciousness could not be accurately assessed. For example, the handgrip strength of coma patients was excluded.

Comment 3: It should be mentioned as a limitation that family members could bring food in from the outside.

(Response)

Thank you for your pointing out. We agree with your pointing out, because it is possible that the patients took in more nutrition than we assessed. In this study, nutrition other than hospital food could not be evaluated. It is possible that the values used in this study are less than the actual amount of nutritional intake. The following statement has been added to the limitation section.

(Change)

(Fifth paragraph of “4. Discussion” section) “The present study has several limitations. First, because it was a retrospective cohort study in a single acute care hospital, we cannot generalize the results, eliminate potential confounders, or prove a causal relationship between energy intake and discharge destination. Second, nurses visually assessed nutritional intake, the main parameter, which may have resulted in measurement errors. Third, the physical functions of patients with impaired consciousness could not be accurately assessed. For example, the handgrip strength of coma patients was excluded. Fourth, we did not record the foods that family members brought from the outside. It is possible that they are taking in more nutrients than we have assessed.

Comment 4: How were nurses trained to evaluate plate waste?

(Response)

We apologize for the lack of detail in the assessment of dietary intake. In previous studies (Ref. 14, 19), training methods were not mentioned for the assessment of dietary intake. In this study, experienced nurse lectured a colleague about dietary intake. For example, young nurses assessed with experienced nurses. The following word have been added to the Methods section.

(Change)

(“2.3. Nutrition intake” section in Materials and Methods) “Based on a previous study [19], we reviewed diet records to quantify the mean daily nutritional intake during the first week after admission to the acute care hospital. Oral intake was measured by the experienced nurses using visual estimation.

Comment 5: Please clarify what is meant by the nurses would measure “grain and dishes intake”?

(Response)

We apologize for the confusing meaning. This means that the nurse looked at the plate after eating, and record the intake of main and side dishes. The following words have been added to the text to aid the reader's understanding.

(Change)

(“2.3. Nutrition intake” section in Materials and Methods) “Based on a previous study [19], we reviewed diet records to quantify the mean daily nutritional intake during the first week after admission to the acute care hospital. Oral intake was measured by the experienced nurses using visual estimation. Visual estimation is commonly used in hospitals to evaluate food intake through the estimation of plate waste [14], and it comprises visually estimating the food present before and after the plate is provided to the patient. After the patient finished the meal, the nurse would register how much food the patient has intake on an 11 point scale of 0–10.

Comment 6: Since rehabilitation time was longer in the other discharge group, was energy expenditure during this activity considered when assessing nutritional needs?

(Response)

We agree that the increasing energy expenditure due to rehabilitation should be taken into account. However, because this study was a retrospective cohort design, it was not possible to adjust energy intake of the other discharge group. In future prospective cohort studies, energy intake should be corrected for energy expenditure due to physical activities.

Comment 7: Assessment in changes in some of these outcomes over time should be included if available (i.e. changes in body weight, FIM, strength, etc). Did caloric inadequacy lead to weight loss in either group?

(Response)

The nutritional intake in the early phase of hospitalization may affect the status at discharge. Therefore, we agree that it is necessary to take into account the changes in body weight and muscle strength over time. However, body weight at discharge was not measured in all patients, and there were many missing data. Therefore, the number of patients who lost body weight due to caloric inadequacy was unknown. Of the patients who could be measured body weight at discharge, those with shorter hospital stays did not lose weight even with energy deficient.

We calculated the FIM gain by subtracting the FIM at admission from the FIM at discharge. When this FIM gain was included in a regression model, FIM gain was not associated with home discharge in this study group (OR = 1.042, 95% CI = 0.979-1.109, p = 0.193).

The following statements have been added to the text to aid the reader's understanding. And, Table 1 and Table 2 have been revised according to the updated results.

(Change)

(“2.4. Outcome measurement” section in Materials and Methods) “. . . The secondary outcome was ADL which was assessed using the Functional Independence Measure (FIM). The FIM score rates 13 motor and 5 cognitive activities on a scale of 1 (complete dependence) to 7 (complete independence). The total FIM score ranges from 18 to 126 points. A high FIM score indicates high activities of daily living [7]. The FIM gain was defined as the FIM at discharge minus the FIM at admission.

(Second paragraph of “4. Discussion” section) “Energy intake in the early phase after onset in acute stroke patients is associated with functional prognosis. In a previous study on acute stroke patients, high energy intake in the early phase after onset resulted in significant improvements in ADL [22]. Kokura et al. measured energy intake for one week after admission and compared it with the basal energy expenditure calculated using the Harris-Benedict equation [19]. They reported that the energy sufficient group had a higher FIM gain at discharge than the energy deficient group, and many patients had improved nutritional status. However, only 39.1% of patients were energy sufficient indicating that many acute stroke patients had inadequate energy intake [19]. Malnutrition is present in 26.4% and 35% of acute stroke patients in the first and second week after admission, respectively [1]. In this study, we did not assess body weight at discharge, so we did not know how many patients lost weight due to energy deficient.

Comment 8: How did actual intake compare to that prescribed by the dietitian? Did each patient see a dietitian at admission?

(Response)

The dietitians used the Harris-Benedict equation to calculate the basal energy expenditure. We agree that it is necessary to confirm that the predicted energy expenditure matches the actual intake. However, the sufficiency rate for the basal energy expenditure was not calculated in this study. In a previous study (Ref. 19), patients with inadequate energy intake compared to basal energy expenditure had less FIM gain and less improvement in nutritional status. In this previous study, 39.1% of acute stroke patients were energy deficient, so it is predicted that there were many energy deficient patients in our study group.

In the hospital where the study was conducted, the dietitians assessed the patient, after the food form was determined by speech therapists. These procedures were conducted within one week of admission.